# Gas Pressure Sensor Based on BDK-Doped Polymer Optical Fiber

**DOI:** 10.3390/mi10110717

**Published:** 2019-10-24

**Authors:** Xin Cheng, Yi Liu, Changyuan Yu

**Affiliations:** 1Photonics Research Centre, Department of Electrical Engineering, The Hong Kong Polytechnic University, Kowloon, Hong Kong SAR, China; chengxin250@163.com; 2Photonics Research Centre, Department of Electronic Information Engineering, The Hong Kong Polytechnic University, Kowloon, Hong Kong SAR, China; changyuan.yu@polyu.edu.hk

**Keywords:** polymers, fiber Bragg gratings, photosensitive material, pressure sensitivity

## Abstract

This paper presents a high sensitivity gas pressure sensor with benzyl-dimethylketal (BDK)-doped polymer optical fiber Bragg grating (POFBG), whose sensitivity is up to 8.12 pm/kPa and 12.12 pm/kPa in positive and negative pressure atmosphere, respectively. The high sensitivity can be explained by its porous chemical structure. The stability and response behavior under air pressure atmosphere has also been investigated. The new understanding of the air pressure response principle and sensitivity difference for the presented sensor can be a worthy reference.

## 1. Introduction

Recently, optical fiber sensors (OFS) have attracted increasing attention because they are small and immune to electromagnetic interference. However, because of the high instinctive Young’s modulus, the deformation of silica fiber under pressure is very small, which results in low sensitivity. Therefore, silica-based OFS is not the best choice in many applications, such as large-strain and high-sensitivity measurements, except when a modified structure is used to enhance sensitivity [1,2,3]. These complex structures are not easy to fabricate and will increase the size of the sensor. Polymer-based optical fiber provides a different solution [4,5] due to its different mechanical and chemical properties. Compared to other POF sensors, the polymer optical fiber Bragg grating (POFBG) sensor has received more attention, especially the POFBG based on polymethyl methacrylate (PMMA) [6]. Due to a lower Young’s modulus of PMMA, which is 25 times lower than that of silica [7], the POFs based on PMMA are more flexible. They exhibit higher failure strain sensitivity and greater elasticity [8]. When they are used for strain, stress, pressure, and temperature monitoring, they show much better performance [9,10,11,12]. In addition, they are biocompatible, which can lead to applications in the biomedical sector [6,13,14], and PMMA can be used for detecting temperature profiles [15], combining with surface plasmon resonance (SPR) technology in bio-sensing applications [16], and tunable dispersion [17]. 

PMMA is a polymer which means that it is macromolecular, and there should be many intervals inside the polymer which allow air or other gas molecules to enter [18]. This process can be used to detect gas pressure. In this paper, we fabricate a pressure sensor based on a homemade POF to measure the air pressure both in the positive and negative range. As there are no commercially available step-index single-mode polymer optical fibers and acknowledging that only a few groups globally are currently fabricating their own POFs using their own procedures, the present characterizations and results will be specific to our fibers but will give valuable information. The results shown in this work are much better compared to previous experimental results. We endeavor to explain the phenomenon and results observed in our experiments based on the specific chemical structure of the PMMA fabricated from our laboratory by our methods.

## 2. Preform Fiber Fabrication and Sensing System 

All the fibers used in this work were drawn from preforms fabricated and named as the “pull-through” method, as mentioned by our group in [6,19], with a core cane inserted into a cladding preform with central air hole. As shown in Figure 1, we made the core preform with a 13-mm initial diameter and drew a cane with 800-μm diameter through it, which consists of benzyl-dimethylketal (BDK) with a concentration of 1 wt% and PMMA. BDK was used instead of benzyl methacrylate (BzMA) which was used in reference [20], to increase the refractive index of the core to make sure that the light can be guided into the core area. Second, we prepared the cladding preform, which was pure PMMA with a diameter of 20-mm, with Teflon wire in the center and then pulled it out after finishing the polymerization in the programmable oven. We obtained a cladding preform with an air hole. Further, the cane was inserted into cladding preform to assemble the full preform. Finally, the preform with a core/cladding ratio of 0.9/20 mm was drawn into 5.6/125 µm fiber using our custom made POF drawing tower. The process is shown in Figure 1. 

From the description above, the cladding preform needed to be heated once before drawing the fiber. However, the method reported by other researchers requires the core material to be poured into the cladding preform and the core material cured by heating at a certain temperature to prepare the full preform. Therefore, it takes twice as long to heat the cladding preform before drawing it. One more temperature treatment process of heating and cooling may significantly influence the mechanical properties and the profile of the reflective index in the core.

To investigate the influence of fiber diameter on the sensitivity, the etching solution, which was prepared from acetone and methanol by volume 3:1, was used to etch the fiber from 125 μm to the desired size. The etching time was 3 mins and 7 mins. The diameters were measured by a microscope, as shown in Figure 2. They were 125 um, 102 µm, and 83 µm, respectively.

For this study, the POFBGs were inscribed by a continuous-wave He-Cd laser at 325 nm (KIMMON IK3501R-G, Kimmon Koha Co., Ltd, Tokyo, Japan), and the phase mask technique was used during the inscription which was used with a 1046.3 nm pitch (Ibsen Photonics A/S, Farum, Denmark). The grating length was 1 cm, and the POF length was 5 cm and connected to a single-mode silica fiber by the gluing method described previously [21]. 

Figure 3a shows the schematic of a gas pressure calibration system. It consists of a pressure producer, a pressure chamber, an interrogator, and a computer. The pressure producer could load a positive or negative pressure separately in the pressure chamber by connecting the producer and chamber using a normal gas pipe. The accurate pressure value in the pressure chamber could be recorded and controlled by a pressure gauge. The POFs with FBG were suspended in the pressure chamber freely. The detecting signal device used in this experiment was an interrogator (SM125-500, MOI, Atlanta, GA, USA), with 2 Hz frequency and 1 pm resolution. The reflected peak and its shift were recorded during the testing. Figure 3b shows the experimental photos of the gas pressure calibration system.

## 3. Experimental Results and Discussions 

At atmospheric pressure, the reflected spectrum of the POF Bragg grating with a diameter of 125 um was measured by using the setup in Figure 3. The reflected spectrum shows a single peak, as shown in Figure 4. It implies that the prepared POF with a 5.6 µm diameter was single-mode fiber. The 3-dB bandwidth was 0.1 nm. When the pressure in the chamber increased, the reflected peak shifted to a longer wavelength. On the contrary, when the pressure in the chamber reduced, the reflected peak shifted to a shorter wavelength. Figure 4 shows the reflected spectrum of the POF Bragg grating at −50 kPa, 0 kPa, and 50 kPa gauge pressure.

Etched fibers were used to conduct the same experiment to investigate the sensitivity of POFBG in the gas pressure chamber. All the data were saved during the testing by 1-minute steps, and the plotted results are shown in Figure 5. The sensitivities of the sensors to the gas pressure were 7.82 pm/kPa, 7.96 pm/kPa, and 8.12 pm/kPa in the positive pressure range corresponding to 125 µm, 102 µm, and 83 µm diameter POFBG sensors, respectively. However, the sensitivities in negative pressure range were different, which were 10.78 pm/kPa, 11.71 pm/kPa, and 12.12 pm/kPa, respectively. 

The results are summarized in Table 1. The sensitivity increased slightly when the diameter of the POF reduced. The sensitivity in the negative pressure atmosphere was higher than that in the positive pressure atmosphere.

The sensitivities were much higher than the results reported by other research groups, as shown in Table 2. Compared to a single-mode silica fiber, the high sensitivity is caused by the low rigidity of the matrix material (PMMA) [13], and the doped BDK in the core improves the sensitivity further. In the present work, the sensitivity of the sensor was as high as that based on the cyclized transparent optical polymer (CYTOP) [22]. However, the FBG [22] was inscribed in the multimode POF. To obtain a single reflected peak, the excited mode has to be controlled exactly. It is difficult in practical application. 

To compare the response time of the sensor in a positive and negative pressure atmosphere, a series of simile experiments were finished, and the different results are shown in Figure 6. The response time in the positive range reduced a little when the pressure increased, whereas the time rose when the negative pressure increased. First, the trend of the sensitivity to different diameter fibers has been explained by K. Bhowmik et.al. [25]. However, they did not report the different response time and sensitivity between positive and negative atmosphere. The polymer was porous, and this porous structure can increase the sensitivity to strain [18]. For applying a positive pressure process, the porous structure in the macromolecule allowed the air to enter the intervals. At the beginning of the process, the air entered the chamber, and the porous PMMA was filled with some air, which is similar to the diffusion process. Pressure was not applied to the POFBG directly and immediately. Therefore, the stabilization time was longer than the latter process (see Figure 6b). After more and more air enters the intervals, the fiber becomes compacted, and then the effective refractive index (RI) should be larger than the normal situation [26]. In addition, after the intervals were filled with more air, the POFBG was sensitive to pressure, so the response time was shorter under larger pressure. For applying the negative pressure process, the progress was totally different. The chains of the PMMA were released and expanded under negative pressure compared to under standard atmosphere, and many air molecules tried to escape from the molecule interval of the PMMA. But the escape progress was different from that under positive pressure. There was no diffusion progress. The gas escaping speed was fast, so the response progress was very fast at the beginning, and the time was much shorter. While the air was escaping, less and less air remained in the intervals, and the Van der Waal’s force was stronger. It needed longer to reach stabilization (see Figure 6a). After some air escaped, the efficient RI was smaller, which produced a shorter wavelength shift. 

To verify the repeatability and reliability, a continual test was implemented. The pressure in the chamber was changed in step. The step was 5 kPa. The step responses of the sensor in negative/positive pressure atmospheres are shown in Figure 7. For each stable pressure, the response of the sensor followed it. 

## 4. Conclusions

To the best of our knowledge, this is the first time the negative pressure measurement based on bare POFBG has been demonstrated. The sensitivity was much higher than previously reported with a maximum of up to 10 times larger than that in positive pressure [20]. The sensitivity of POFBG under a negative atmosphere was higher than that in positive. This is due to the porous chemical structure of PMMA, which makes sense when viewing the results. In addition, stability and response time were investigated under two different air atmospheres. The reasonable explanation from the perspective of a porous chemical structure can be useful in understanding the response principle. The sensitivity under lower pressure range could be detected based on the presented sensor, which has potential in the small pressure change detection area, such as subcutaneous tissue pressure measurement [27].

## Figures and Tables

**Figure 1 micromachines-10-00717-f001:**
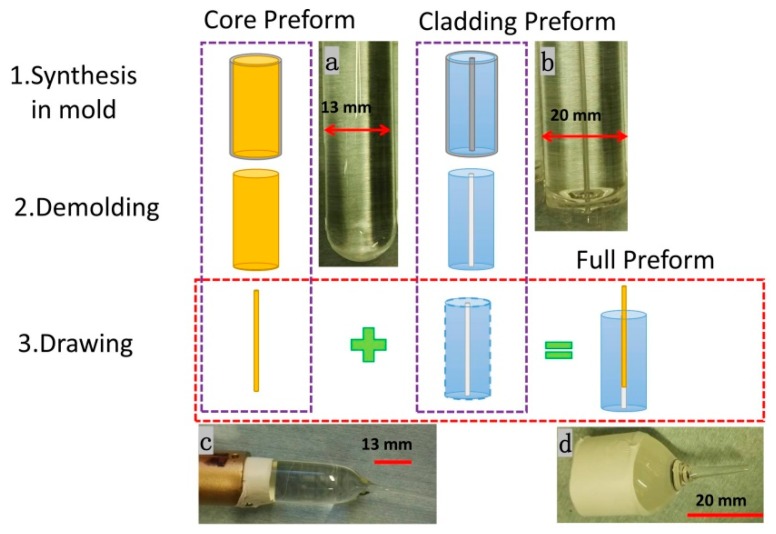
Preparation process of the polymer optical fiber (POF) preform; Subfigure (**a**–**d**) are the photos of the core preform, cladding preform with a hole, the drawn core cane, and the full preform.

**Figure 2 micromachines-10-00717-f002:**
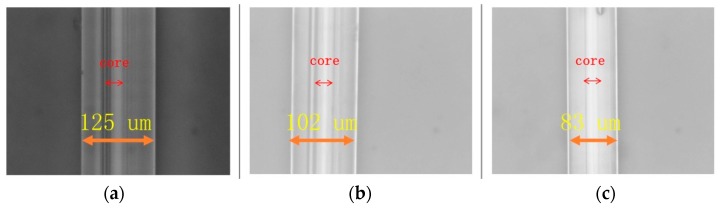
Microscope images of the fibers before/after etching. (**a**) Freshly prepared sample; (**b**) 3 min etching; (**c**) 7 min etching

**Figure 3 micromachines-10-00717-f003:**
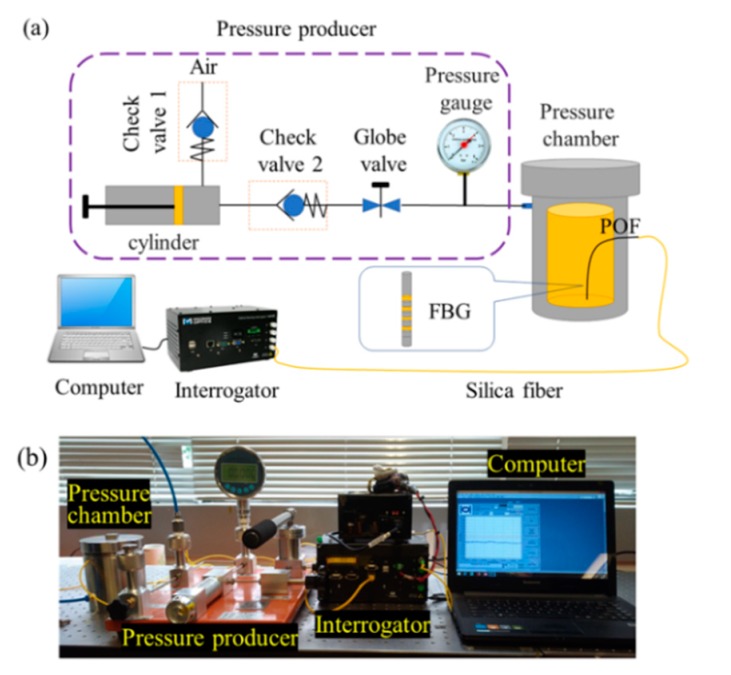
(**a**) The schematic diagram and (**b**) the experimental photo of the gas pressure calibration system.

**Figure 4 micromachines-10-00717-f004:**
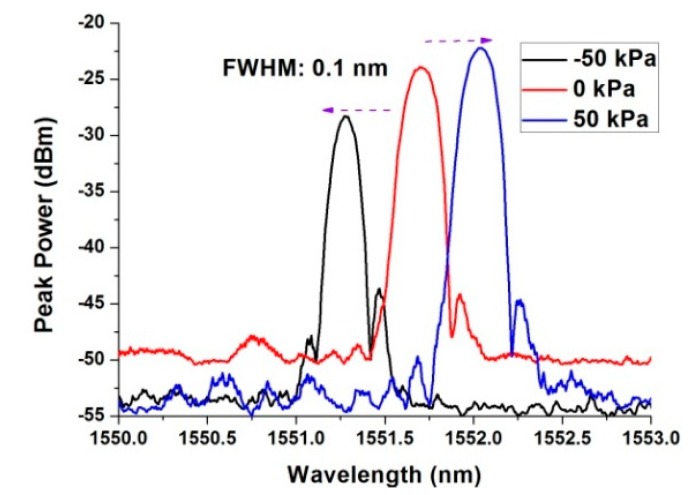
The reflected spectrum of the POF Bragg grating.

**Figure 5 micromachines-10-00717-f005:**
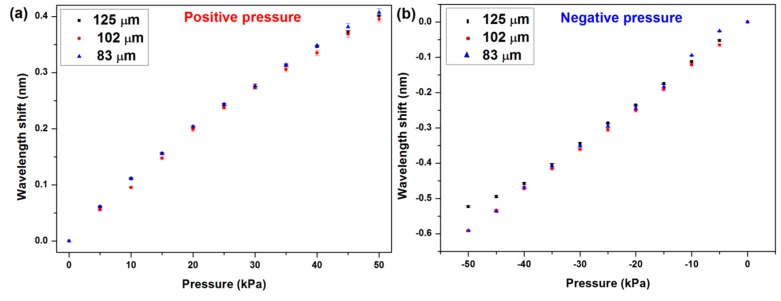
(**a**) Positive pressure sensitivities and (**b**) negative pressure sensitivities of the sensors with different diameters.

**Figure 6 micromachines-10-00717-f006:**
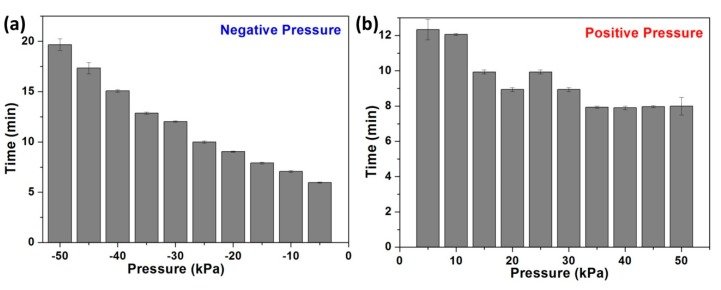
Evaluation of response time in two different pressure atmospheres—negative (**a**) and positive (**b**).

**Figure 7 micromachines-10-00717-f007:**
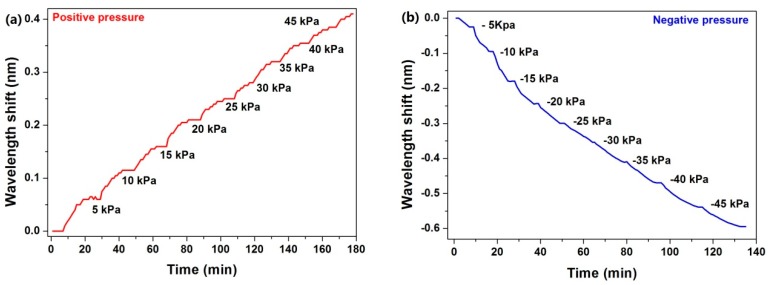
Step response of the sensor in negative/positive pressure atmosphere—positive (**a**) and negative (**b**).

**Table 1 micromachines-10-00717-t001:** Sensitivity of different diameter fiber Bragg grating (FBG) sensors under positive and negative pressure.

Diameter	Sensitivity	
	Positive Pressure	Negative Pressure
125 µm	7.82 pm/kPa	10.79 pm/kPa
102 µm	7.97 pm/kPa	11.71 pm/kPa
83 µm	8.12 pm/kPa	12.12 pm/kPa

**Table 2 micromachines-10-00717-t002:** The different sensitivities reported by research groups.

Fiber	Sensitivity	Range	Reference
Single-mode silica fiber	3.04 pm/MPa	0–70 MPa	[23]
BzMA-doped core with PMMA cladding SMPOF	200~750 pm/MPa	0–1000 kPa	[20]
BDK-doped core with PMMA cladding SMPOF	~7000 pm/MPa	0–50 kPa	Present work
CYTOP multimode POF	7710–8510 pm/MPa	10–1500 kPa	[24]

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
