# Peer review of "Gas Pressure Sensor Based on BDK-Doped Polymer Optical Fiber"

_micromachines, 2019, doi:10.3390/mi10110717_

Round 1

Reviewer 1 Report

In this paper, the authors reported a high sensitivity gas pressure sensors with BDK doped POFBG for both positive and negative atmosphere, here, for negative atmosphere is more interesting. The article is very well written, clear and concise and suitable for the scope of the journal. Only suggestions are supplied:

Please give more detail for the picture in Fig. 1 such as diameter of the preform. Also “Preform” not “preform” in the Fig.1 sentence. Also for Fig.2 , please give more detail such as the location of the fiber core. Please check reference 15. Actually, a lot of novel applications based on POF reported recently, such as SPR detect, thermal profile detect and dispersion compensation. Some of these are worth to introduce in the introduction part.

Author Response

Firstly, Thanks for the reviewer's work and I replied to the questions in the below content. And I also revised the manuscript according to the response. 

The diameter of the cladding preform is 20mm, the core preform is 13 mm diameter and the “Preform” had been changed in Fig.1 The core area is clear under the microscope, and it is in the center part. And the core area had been marked in Fig.2. More reference papers had been added in the introduction part. (in Line 31,32)

Reviewer 2 Report

The authors propose an optical fiber sensor with BDK-doped Polymer optical fiber Bragg grating (POFBG) for gas pressure measurement. The fabrication of the FOFBG, measurement design, and the experimental results are clearly presented. However, the degree of novelty/impact in this manuscript is not very significant. I don’t recommend this paper to be published in the Micromachines unless authors explain some more unique advantages of this proposed sensor and improve the manuscript.

Besides, some of my concerns are illustrated in the following:

In the Introduction, I don’t agree the silica fiber sensor cannot be used for high-sensitivity measurements unless the authors propose examples or evidence.   Lack of the detail of the phase mask for fabricating the optical fiber grating. Lack of the producer information (company, city, and country) and the specifications of the interrogator. Is MOI a company name? What is the resolution? How do authors define “negative pressure”? For example, -50 kp means what? For a vacuum, the pressure theoretically is zero. It means no pressure. So, how come to get a minus value? I guess the authors wanted to express the cases of pressurizing and pressure releasing. Am I right? How many times for each pressure test (including different diameters of POFBG, positive pressure, negative pressure)?   How were the sensitivities (e.g. 7.82 pm/kPa, 7.96 pm/kPa... ) calculated? I think the resolution of the interrogator should also be taken into account. For example, if the resolution is bigger than 10 pm, how come the sensitivity of the sensor could be less than 10? The response times are in the range of 5 to 20 minutes. That means the measurement can not be real-time. Therefore the practicability of this POFBG sensor is doubtful.

    8. Figure 6 just shows 4 error-bars. Can the authors explain why?

Author Response

Firstly, I want to state that I appreciated the work from the Reviewer and the comments are very useful to improve the manuscript.  I replied to all comments and revised the manuscript according to the response.

Round 2

Reviewer 2 Report

First, a standard senor should be qualified by repeatable measurements whether the measured values are increasing or decreasing. The main problems in this study are in the following.

The unlike express of air pressure. In engineering, the pressure of an absolute vacuum is zero. So, the “negative pressure” is nonsense. Whether pressurizing or reducing pressure, there is only one absolute pressure value. In this study, the so-called “positive” and “negative” pressures have different measurement data, which is unlikely. The truth in this study is that the measurements during pressuring and reducing pressure have different response times. The authors declare that their optical fiber sensor can measure negative pressure, which is nonsense. The correct approach should be measuring the air pressure by starting from low pressure to the highest pressure, and then from the highest pressure to the lowest pressure. And then repeat it some times to verify the reproducibility. The response times are in the range of 5 to 20 minutes. But the authors didn’t explain how long they recorded new data. In addition, if the reaction times are variable, then how does the author confirm that the record times are correct?

According to the above problems, my previous suggestion was a major revise before publishing. Besides, I still have some concerns in the following:

In the Introduction, still no reference for that the silica fiber sensor cannot be used for high-sensitivity measurements.    The phase mask for fabricating the optical fiber grating and the interrogator should be indicated with the company/city/country. In Fig. 1, the drawings need showing dimension; c & d need showing scale bars. In 3. Experimental results and discussions, “whereas applying negative pressure into the chamber” is an unlikely expressing. It should be “releasing pressure from the chamber”. In line #127, “a litter bit” should be “a little bit”.

Author Response

Dear Reviewer,

Firstly, thank you for your valuable comments. 

The authors read your comments carefully and answer your questions one by one. The response letter is attached and Please see the attachment.

The authors revised the text of the manuscript according to the response letter. The revised parts were highlight by red color. 

Round 3

Reviewer 2 Report

no comment for the authors